# Association of severe mental illness and septic shock case fatality rate in patients admitted to the intensive care unit: A national population-based cohort study

Ines Lakbar[1,2], Marc Leone[2], Vanessa Pauly[1], Veronica Orleans[1], Kossi Josue Srougbo[1], Sambou Diao[1], Pierre-Michel Llorca[3,4], Marco Solmi[5,6,7,8,9], Christoph U. Correll[9,10,11], Sara Fernandes[1], Jean-Louis Vincent[12], Laurent Boyer[1,3]*, Guillaume Fond[1,3]

1 AP-HM, Aix-Marseille Univ, CEReSS, Health Service Research and Quality of Life Centre, School of Medicine—La Timone Medical, Marseille, France, 2 Aix-Marseille University, AP-HM, North Hospital, Department of Anaesthesia and Intensive Care Unit, Marseille, France, 3 FondaMental Fondation, Créteil, France, 4 University Clermont Auvergne, CMP-B CHU, CNRS, Clermont Auvergne INP, Institut Pascal, Clermont-Ferrand, France, 5 Department of Psychiatry, University of Ottawa, Ontario, Canada, 6 Department of Mental Health, Ottawa Hospital, Ontario, Canada, 7 Ottawa Hospital Research Institute (OHRI), Clinical Epidemiology Program, University of Ottawa, Ottawa, Ontario, Canada, 8 School of Epidemiology and Public Health, Faculty of Medicine, University of Ottawa, Ottawa, Canada, 9 Charité-Universitätsmedizin Berlin, corporate member of Freie Universität Berlin, Humboldt-Universität zu Berlin, and Berlin Institute of Health, Department of Child and Adolescent Psychiatry, Berlin, Germany, 10 The Zucker Hillside Hospital, Department of Psychiatry, Glen Oaks, New York, United States of America, 11 The Donald and Barbara Zucker School of Medicine at Hofstra/Northwell, Department of Psychiatry and Molecular Medicine, Hempstead, New York, United States of America, 12 Department of Intensive Care, Erasme Hospital, Université libre de Bruxelles, Brussels, Belgium

* laurent.boyer@ap-hm.fr

## Abstract

### Background

Patients with severe mental illness (SMI) (i.e., schizophrenia, bipolar disorder, or major depressive disorder) have been reported to have excess mortality rates from infection compared to patients without SMI, but whether SMI is associated with higher or lower case fatality rates (CFRs) among infected patients remains unclear. The primary objective was to compare the 90-day CFR in septic shock patients with and without SMI admitted to the intensive care unit (ICU), after adjusting for social disadvantage and physical health comorbidity.

### Methods and findings

We conducted a nationwide, population-based cohort study of all adult patients with septic shock admitted to the ICU in France between January 1, 2014, and December 31, 2018, using the French national hospital database. We matched (within hospitals) in a ratio of 1:up to 4 patients with and without SMI (matched-controls) for age (5 years range), sex, degree of social deprivation, and year of hospitalization. Cox regression models were conducted with adjustment for smoking, alcohol and other substance addiction, overweight or obesity,

**Data Availability Statement:** The data are not freely available, they are accessible on a French national platform (ATIH). No personal data can be extracted from this platform (even for those who have access), only aggregated results can be

extracted. All the information can be asked at this address: https://www.atih.sante.fr/nous-contacter.

**Funding:** The authors received no specific funding for this work.

**Competing interests:** I have read the journal's policy and the authors of this manuscript have the following competing interests: LB has received honoraria/has been a consultant for Lundbeck. PML participated in advisory boards for, received speaker's honoraria and received consultation fees over the last 3 years from Eisai, Ethypharm, Janssen, Lundbeck, MSD, Neuraxpharm, Novartis, Otsuka, Roche, Rovi; Member of the Executive Committee of the Fondation FondaMental. GF has received honoraria/has been a consultant for Lundbeck. MS has received honoraria/has been a consultant for Angelini, Lundbeck, Otsuka. CUC has been a consultant and/or advisor to or has received honoraria from AbbVie, Acadia, Alkermes, Allergan, Angelini, Aristo, Boehringer-Ingelheim, Cardio Diagnostics, Cerevel, CNX Therapeutics, Compass Pathways, Damitsa, Gedeon Richter, Hikma, Holmusk, IntraCellular Therapies, Janssen/J&J, Karuna, LB Pharma, Lundbeck, MedAvante-ProPhase, MedInCell, Merck, Mindpax, Mitsubishi Tanabe Pharma, Mylan, Neurocrine, Newron, Noven, Otsuka, Pharmabrain, PPD Biotech, Recordati, Relmada, Reviva, Rovi, Seqirus, SK Life Science, Sunovion, Sun Pharma, Supernus, Takeda, Teva, and Viatris. He provided expert testimony for Janssen and Otsuka. He served on a Data Safety Monitoring Board for Lundbeck, Relmada, Reviva, Rovi, Supernus, and Teva. He has received grant support from Janssen and Takeda. He received royalties from UpToDate and is also a stock option holder of Cardio Diagnostics, Mindpax, and LB Pharma. ML has received honoraria from Gilead, AOP Pharma, Ambu, LFB and Viatris for consulting.

**Abbreviations:** aHR, adjusted hazard ratio; CCAM, Classification Commune des Actes Médicaux; CFR, case fatality rate; CI, confidence interval; COVID-19, Coronavirus Disease 2019; DRG, diagnosis-related group; FIASMA, functional inhibition of acid sphingomyelinase; HR, hazard ratio; ICD-10, 10th revision of the International Classification of Diseases; ICU, intensive care unit; IFN-γ, interferon-gamma; IL, interleukin; PMSI, Programme de Médicalisation des Systèmes d'Information; SAPS II, Simplified Acute Physiology Score II; SD, standardized difference; SMI, severe mental illness.

Charlson comorbidity index, presence of trauma, surgical intervention, Simplified Acute Physiology Score II score, organ failures, source of hospital admission (home, transfer from other hospital ward), and the length of time between hospital admission and ICU admission. The primary outcome was 90-day CFR. Secondary outcomes were 30- and 365-day CFRs, and clinical profiles of patients.

A total of 187,587 adult patients with septic shock admitted to the ICU were identified, including 3,812 with schizophrenia, 2,258 with bipolar disorder, and 5,246 with major depressive disorder. Compared to matched controls, the 90-day CFR was significantly lower in patients with schizophrenia (1,052/3,269 = 32.2% versus 5,000/10,894 = 45.5%; adjusted hazard ratio (aHR) = 0.70, 95% confidence interval (CI) 0.65,0.75, $p < 0.001$), bipolar disorder (632/1,923 = 32.9% versus 2,854/6,303 = 45.3%; aHR = 0.70, 95% CI = 0.63,0.76, $p < 0.001$), and major depressive disorder (1,834/4,432 = 41.4% versus 6,798/14,452 = 47.1%; aHR = 0.85, 95% CI = 0.81,0.90, $p < 0.001$). Study limitations include inability to capture deaths occurring outside hospital, lack of data on processes of care, and problems associated with missing data and miscoding in medico-administrative databases.

## Conclusions

Our findings suggest that, after adjusting for social disadvantage and physical health comorbidity, there are improved septic shock outcome in patients with SMI compared to patients without. This finding may be the result of different immunological profiles and exposures to psychotropic medications, which should be further explored.

## Author summary

### Why was this study done?

- Patients with severe mental illness (SMI) have been reported to have excess mortality from sepsis (number of deaths due to sepsis in the whole population).

- Whether SMI is associated with higher or lower sepsis-associated case fatality remains unclear (number of deaths due to sepsis in the population with sepsis).

- No study has determined whether SMI is associated with excess case fatality in patients with septic shock, the most severe form of sepsis when accounting for the most relevant confounding variables.

### What did the researchers do and find?

- In this nationwide, population-based cohort study, we compared 30-, 90-, and 365-day case fatality rates (CFRs) in septic shock patients with and without SMI admitted to the intensive care unit (ICU).

- We identified 187,587 adult patients with septic shock admitted to the ICU, including 3,812 with schizophrenia, 2,258 with bipolar disorder, and 5,246 with major depressive disorder.

- The 30-, 90-, and 365-day CFRs were lower in patients with SMI than in patients without SMI after controlling for multiple potential confounding factors (using intrahospital matching and adjustments for multiple comorbidities and illness severity) and addressing potential biases not considered in previous studies.

## What do these findings mean?

- Our findings suggest improved septic shock outcomes in patients with SMI compared to patients without.

- Our findings also suggest that the excess mortality from sepsis is due to an increased risk of sepsis/infection among patients with SMI, but not due to increase case fatality among septic patients.

- This finding may be the result of different immunological profiles and exposures to psychotropic medications, a hypothesis that needs to be confirmed in future studies.

## Introduction

Data have consistently indicated that individuals with severe mental illness (SMI) (i.e., schizophrenia, bipolar disorder, or major depressive disorder) are at higher risk of premature mortality than the general population [1,2]. This is mainly attributed to higher rates of physical disease, social disadvantage, unhealthy lifestyle behaviors, and inadequate healthcare in patients with SMI [3–6]. Among somatic diseases, infections are disproportionately more frequent in patients with SMI than in the general population, representing a potentially avoidable contributor to early death [2,7,8]. In a meta-analysis, patients with SMI were reported to have higher mortality rates from infection than the general population [2].

Whether SMI is associated with higher or lower infection-associated case fatality (i.e., the proportion of persons with infection who die from that infection [9]) compared with the general population is unclear. Sepsis (i.e., infection-associated organ dysfunction) is one of the leading causes of death around the world [10], with in-hospital case fatality rates (CFRs) as high as 40% in septic shock, the most severe form of sepsis [11]. Few studies have reported data on sepsis-associated CFR in patients with SMI, showing conflicting results: 2 studies reported higher CFR [12,13] and 4 studies reported lower CFR [14–17]. These latter 4 studies performed additional adjustments but omitted important confounding factors, such as overweight or obesity status, severity of sepsis, and type of hospital. Presence of overweight/obesity may represent a protective factor [18] and is more prevalent in patients with SMI than in the general population [19]. Because of the bias associated with variability and subjectivity of sepsis diagnosis [20–22], there is a need to adjust for severity of illness using an appropriate scoring system [23]. Finally, patients with SMI are more often hospitalized at university hospitals [24–26], which are characterized by higher sepsis case volumes known to be associated with better survival [27], than in smaller hospitals [24,25]. Patient matching within a hospital has been advocated to control best for facility confounders [28].

To the best of our knowledge, to date, no study has determined whether SMI is associated with excess CFR in patients with septic shock after accounting for the most relevant confounding variables. To address this issue, we conducted a nationwide, population-based cohort study using the French national hospital database. The primary objective was to compare

90-day CFRs in septic shock patients with and without SMI admitted to the intensive care unit (ICU), after adjusting for social disadvantage and physical health comorbidity. Secondary objectives were to compare 30- and 365-day CFRs and clinical profiles in septic shock patients with and without SMI. We hypothesized that patients with SMI would have a higher septic shock CFR than patients without SMI.

## Methods

### Study design, sources, and population

In this nationwide, population-based cohort study, we used data from the Programme de Médicalisation des Systèmes d'Information (PMSI database), the French national hospital database in which administrative and medical data are systematically collected for acute (PMSI-MCO) and psychiatric (PMSI-PSY) hospitalizations. The PMSI database is based on diagnosis-related groups (DRGs), with all diagnoses coded according to the 10th revision of the International Classification of Diseases (ICD-10) and using procedural codes from the Classification Commune des Actes Médicaux (CCAM). The PMSI database is used to determine financial resource use and is frequently and carefully verified by its producer as well as the paying party, with possible financial and legal consequences. Data from the PMSI database are anonymized and can be reused for research purposes. A unique anonymous identifier enables different inpatient stays of individual patients to be linked. The study was submitted to the French National Data Protection Commission (N° 2203797) for ethical approval. This manuscript follows the Strengthening the Reporting of Observational Studies in Epidemiology (STROBE) guidelines [29] (S1 STROBE Checklist).

We included all hospital admissions between January 1, 2014, and December 31, 2018, using the following criteria: aged 18 years or older, admitted to the ICU, had a diagnosis of septic shock (ICD-10 code = R572 or a combination of codes corresponding to a severe infection associated with the use of vasopressors). We limited inclusion to patients with an ICU length of stay of at least 48 hours, unless the patient died within 48 hours, in order to avoid overestimating diagnoses of septic shock. Although the coding of septic shock has been strictly regulated since the DRG system was introduced in France, we cannot exclude overcoding due to the high tariff associated with the codes, especially for short stays in the ICU. Indeed, the length of stay for patients with septic shock is about 7 days (IQR 3 to 14 days) [30]. We therefore considered the first quartile ($< = 2$ days) to be a credible threshold below which the probability of having septic shock was low (excluding patients who died within these 48 hours).

### Outcomes

The primary outcome was 90-day CFR (i.e., deaths per 100 cases of septic shock, percentage). Secondary outcomes were 30- and 365-day CFRs and the clinical profiles of patients.

### Collected data

We collected the following sociodemographic data: age, sex, and degree of social deprivation (least deprived, less deprived, more deprived, most deprived according to quartiles) based on 4 socioeconomic ecological variables—the proportion who had graduated from high school, median household income, the percentage of blue-collar workers, and the unemployment rate [31]. We also collected data on comorbidities (overweight or obesity, addiction [smoking, alcohol, and other substances], Charlson Comorbidity Index (0, 1 to 2, $\geq 3$ [32]); presence of trauma; surgical intervention; Simplified Acute Physiology Score II (SAPS II) at ICU admission; source of infection and identified pathogens; the type of organ failure (respiratory, renal,

neurologic, cardiovascular, hematologic, metabolic); and use of supportive therapies (cardio-pulmonary resuscitation, invasive mechanical ventilation, renal replacement therapy, transfu-sion). Characteristics of the stay were noted, including the source of hospital admission (i.e., where the patient came from [home, transfer from other hospital ward]), the length of time between hospital admission and ICU admission, and durations of ICU and hospital stay; char-acteristics of the hospital were also recorded (academic, general public, and private).

## Exposures

For the purpose of this study, we defined 6 groups: 3 groups with SMI, which included patients with a diagnosis of schizophrenia (ICD-10 codes F20*, F22*, or F25*), bipolar disorder (ICD-10 codes F30*, F31*), or major depressive disorder (ICD-10 codes F33*), and 3 matched groups without SMI (controls). The control groups were created by matching for age (5-year range), sex, degree of social deprivation, and year of hospitalization in a ratio of 1:up to 4 patients with and without SMI within a hospital (to control for confounders at a hospital level). In patients with dual diagnoses, those with codes for schizophrenia and bipolar disorder or major depressive disorder were classified in the schizophrenia group, and those with codes for bipolar disorder and major depressive disorder were classified as bipolar disorder. There was therefore no overlap across the groups.

## Statistical analysis

The patients' characteristics are presented as counts (percentages) and medians (interquartile ranges) for categorical and continuous variables, respectively. CFR was calculated at 30, 90, and 365 days using the total number of patients admitted to the ICU with septic shock as the denominator.

   Standardized differences were used to compare patients with and without SMI using weights to normalize the distribution of patients. An absolute standardized difference (SD) of ⬚0.20 was chosen to indicate a negligible difference in the mean or prevalence of a variable between groups [33]. The SD helps to understand the magnitude of the differences found, in addition to statistical significance, which examines whether the findings are likely to be due to chance [34].

   To study the association between each SMI and outcome, the Kaplan–Meier method and the log-rank statistic were used to estimate and compare the cumulative death rates. Hazard ratios (HRs) and 95% confidence intervals (95% CIs) were estimated using Cox survival mod-els with a robust variance estimator to account for clustering within matched pairs. Two mod-els were developed for each outcome. Model 1 included SMI only (no adjustment). Model 2 included SMI with additional covariates of smoking, alcohol, and other substance addiction (yes versus no), overweight or obesity (yes versus no), the Charlson comorbidity index (0, 1 to 2, ≥3), presence of trauma (yes versus no), surgical intervention (yes versus no), SAPS II score (modified, without age), organ failures (yes versus no for each of respiratory, renal, neurologic, cardiovascular, hematologic, metabolic, hepatic), the source of hospital admission (home, transfer from other hospital ward), and time between hospital admission and ICU admission (≤1 versus > 1 day). The covariates were selected a priori on the basis of clinical relevance or the results of bivariate outcomes analyses (SD > 0.2). Interactions with SMI were investigated, but associations were negligible. Several sensitivity analyses were performed: model S1 (model 2 with the 17 Charlson comorbidities instead of the Charlson comorbidity index), model S2 (model 2 with infected organs instead of organ failures), model S3 (model 2 with ICU support-ive therapies instead of organ failures), model S4 (model 2 with the nature of isolated patho-gens), and model S5 on the whole cohort (without matching process) using the same variables

as in model 2 and matching variables to consider residual bias from incomplete matching of controls to the respective SMI group.

The proportional-hazards assumption for the Cox models was investigated and confirmed graphically through survival functions over time. A $p < 0.05$ was considered significant. Data management and analyses were performed using the SAS software. Cox regression analyses were performed using the PROC PHREG in SAS.

## Results

The database included a total of 187,587 patients with septic shock (flow chart, **Fig 1**). The main sociodemographic data of the patients are shown in **Table 1**. The mean age was 67.1 (±14.3) years and 63.8% were men. A majority of patients (106,941 patients [57.0%]) were socially deprived and most patients (167,738 patients [89.4%]) were hospitalized in public hospitals. Among the 187,587 patients, 3,812 had schizophrenia (2.0%), 2,258 had bipolar disorder (1.2%), and 5,246 had major depressive disorder (2.8%). A total of 3,269 patients with schizophrenia, 1,923 patients with bipolar disorder, and 4,432 patients with major depressive disorder were matched with 10,894, 6,303, and 14,452 controls, respectively.

### Comparison of CFRs in septic shock patients with and without SMI

Compared to matched controls, the 90-day CFR was significantly lower in patients with schizophrenia (1,052/3,269 = 32.2% versus 5,000/10,894 = 45.5%; adjusted HR (aHR) = 0.70, 95% CI 0.65,0.75, $p < 0.001$), bipolar disorder (632/1,923 = 32.9% versus 2,854/6,303 = 45.3%; aHR = 0.70, 95% CI = 0.63,0.76, $p < 0.001$), and major depressive disorder (1,834/4,432 = 41.4% versus 6,798/14,452 = 47.1%; aHR = 0.85, 95% CI = 0.81,0.90, $p < 0.001$) (**Tables 2 and 3**).

The 30-day and 365-day CFRs were also significantly lower in patients with schizophrenia, bipolar disorder, and major depressive disorder than in matched controls. The sensitivity analyses reported similar findings for 30-, 90-, and 365-day CFRs (**S1, S2, and S3 Figs**). **S4 Fig** shows the survival curves in the different groups at 1 year.

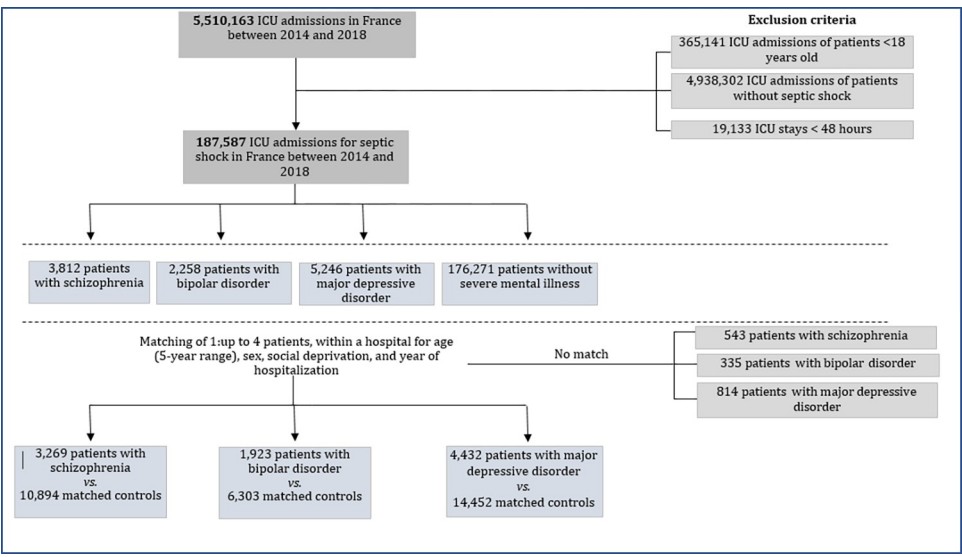

**Fig 1. Flow chart of the patients admitted to the intensive care unit (ICU) with septic shock during the study period.**

**Table 1. Sociodemographic and hospital characteristics in the different groups, and crude septic shock case fatality of patients before matching.**

| | All | Patients with schizophrenia | Patients with bipolar disorder | Patients with major depressive disorder | Patients without SMI | SD† | p-value† | SD‡ | p-value‡ | SDƒ | p-valueƒ |
|---|---|---|---|---|---|---|---|---|---|---|---|
| N | 187,587 | 3,812 | 2,258 | 5,246 | 176,271 | - | | - | | - | |
| Age–year | | | | | | | | | | | |
| Mean ± SD [95% CI] | 67.1 ± 14.3 [67.1–67.2] | 58.2 ± 14.3 [57.8–58.7] | 62.6 ± 13.0 [62.1–63.2] | 63.6 ± 14.2 [63.2–63.9] | 67.5 ± 14.2 [67.4–67.6] | −0.65 | <0.001 | −0.36 | <0.001 | −0.27 | <0.001 |
| Distribution–n (%) [95% CI] | | | | | | | <0.001 | | <0.001 | | <0.001 |
| 18–44 | 13,438 (7.2%) [7.0–7.3] | 623 (16.3%) [15.2–17.5] | 209 (9.3%) [8.1–10.5] | 475 (9.1%) [8.3–9.8] | 12,131 (6.9%) [6.8–7.0] | 0.30 | | 0.09 | | 0.08 | |
| 45–64 | 58,131 (31.0%) [30.8–31.2] | 1,886 (49.5%) [47.9–51.0] | 965 (42.7%) [40.7–44.8] | 2,205 (42.0%) [40.7–43.4] | 53,075 (30.1%) [29.9–30.3] | 0.40 | | 0.26 | | 0.25 | |
| 65–75 | 56,917 (30.3%) [30.1–30.5] | 862 (22.6%) [21.3–23.9] | 718 (31.8%) [29.9–33.7] | 1,373 (26.2%) [25.0–27.4] | 53,964 (30.6%) [30.4–30.8] | −0.18 | | 0.03 | | −0.10 | |
| >75 | 59,101 (31.5%) [31.3–31.7] | 441 (11.6%) [10.5–12.6] | 366 (16.2%) [14.7–17.7] | 1,193 (22.7%) [21.6–23.9] | 57,101 (32.4%) [32.2–32.6] | −0.52 | | −0.38 | | −0.22 | |
| Age at death–year | | | | | | | | | | | |
| Mean ± SD [95% CI] | 69.9 ± 13.0 [69.8–70.0] | 62.1 ± 13.6 [61.3–62.9] | 66.7 ± 12.2 [65.8–67.6] | 66.8 ± 13.1 [66.2–67.4] | 70.1 ± 12.9 [70.0–70.2] | −0.60 | <0.001 | −0.27 | <0.001 | −0.25 | <0.001 |
| Sex–n (%) [95% CI] | | | | | | | | | | | |
| Women | 67,816 (36.2%) [35.9–36.4] | 1,450 (38.0%) [36.5–39.6] | 1,214 (53.8%) [51.7–55.8] | 2,819 (53.7%) [52.4–55.1] | 62,333 (35.4%) [35.1–35.6] | −0.06 | <0.001 | −0.38 | <0.001 | −0.38 | <0.001 |
| Social deprivation, n (%) [95% CI] | | | | | | | <0.001 | | <0.001 | | 0.714 |
| Least deprived | 51,939 (27.7%) [27.5–27.9] | 1,194 (31.3%) [29.8–32.7] | 704 (31.2%) [29.3–33.1] | 1,447 (27.6%) [26.4–28.8] | 48,594 (27.6%) [27.4–27.8] | 0.08 | | 0.08 | | 0.00 | |
| Less deprived | 28,707 (15.3%) [15.1–15.5] | 599 (15.7%) [14.6–16.9] | 360 (15.9%) [14.4–17.5] | 830 (15.8%) [14.8–16.8] | 26,918 (15.3%) [15.1–15.4] | 0.01 | | 0.02 | | 0.02 | |
| More deprived | 61,575 (32.8%) [32.6–33.0] | 1,179 (30.9%) [29.5–32.4] | 724 (32.1%) [30.1–34.0] | 1,701 (32.4%) [31.2–33.7] | 57,971 (32.9%) [32.7–33.1] | −0.04 | | −0.02 | | −0.01 | |
| Most deprived | 45,366 (24.2%) [24.0–24.4] | 840 (22.0%) [20.7–23.4] | 470 (20.8%) [19.1–22.5] | 1,268 (24.2%) [23.0–25.3] | 42,788 (24.3%) [24.1–24.5] | −0.05 | | −0.08 | | −0.00 | |
| Year, n (%) [95% CI] | | | | | | | 0.832 | | 0.363 | | 0.272 |
| 2014 | 34,728 (18.5%) [18.3–18.7] | 682 (17.9%) [16.7–19.1] | 420 (18.6%) [17.0–20.2] | 994 (19.0%) [17.9–20.0] | 32,632 (18.5%) [18.3–18.7] | −0.02 | | 0.00 | | 0.01 | |

*(Continued)*

**Table 1.** (Continued)

| | All | Patients with schizophrenia | Patients with bipolar disorder | Patients with major depressive disorder | Patients without SMI | SD† | p-value† | SD‡ | p-value‡ | SDƒ | p-valueƒ |
|---|---|---|---|---|---|---|---|---|---|---|---|
| 2015 | 37,114 (19.8%) [19.6–20.0] | 768 (20.2%) [18.9–21.4] | 459 (20.3%) [18.7–22.0] | 1,067 (20.3%) [19.3–21.4] | 34,820 (19.8%) [19.6–19.9] | 0.01 | | 0.01 | | 0.01 | |
| 2016 | 37,857 (20.2%) [20.0–20.4] | 768 (20.2%) [18.9–21.4] | 484 (21.4%) [19.7–23.1] | 1,081 (20.6%) [19.5–21.7] | 35,524 (20.2%) [20.0–20.3] | −0.00 | | 0.03 | | 0.01 | |
| 2017 | 38,238 (20.4%) [20.2–20.6] | 794 (20.8%) [19.5–22.1] | 434 (19.2%) [17.6–20.8] | 1,050 (20.0%) [18.9–21.1] | 35,960 (20.4%) [20.0–20.3] | 0.01 | | −0.03 | | −0.01 | |
| 2018 | 39,650 (21.1%) [21.0–21.3] | 800 (21.0%) [19.7–22.3] | 461 (20.4%) [18.8–22.1] | 1,054 (20.1%) [19.0–21.2] | 37,335 (21.2%) [21.0–21.4] | −0.00 | | −0.02 | | −0.03 | |
| Hospital characteristics, n (%) [95% CI] | | | | | | | <**0.001** | | <**0.001** | | <**0.001** |
| Academic | 63,230 (33.7%) [33.5–33.9] | 1,568 (41.1%) [39.6–42.7] | 857 (38.0%) [35.9–40.0] | 1,928 (36.8%) [35.4–38.1]) | 58,877 (33.4%) [33.2–33.6] | 0.16 | | 0.10 | | 0.07 | |
| Other public hospital | 104,508 (55.7%) [55.5–55.9] | 2,067 (54.2%) [52.6–55.8] | 1253 (55.5%) [53.4–57.5] | 2,980 (56.8%) [55.5–58.1] | 98,208 (55.7%) [55.5–55.9] | −0.03 | | −0.00 | | 0.02 | |
| Private | 19,849 (10.6%) [10.4–10.7] | 177 (4.6%) [4.0–5.3] | 148 (6.6%) [5.5–7.6] | 338 (6.4%) [5.8–7.1] | 19,186 (10.9%) [10.7–11.0] | **−0.23** | | −0.15 | | −0.16 | |
| Crude case fatality, n (%) [95% CI] | | | | | | | | | | | |
| 30-day case fatality | 75,531 (40.3%) [40.0–40.5] | 923 (24.2%) [22.9–25.6] | 563 (24.9%) [23.1–26.7] | 1,689 (32.2%) [30.9–33.5] | 72,356 (41.1%) [40.8–41.3] | **−0.37** | <**0.001** | **−0.35** | <**0.001** | −0.18 | <**0.001** |
| 90-day case fatality | 91,476 (48.8%) [48.5–49.0] | 1,207 (31.7%) [30.2–33.1] | 730 (32.3%) [30.4–34.3] | 2,134 (40.7%) [39.3–42.0] | 87,405 (49.6%) [49.4–49.8] | **−0.37** | <**0.001** | **−0.36** | <**0.001** | −0.18 | <**0.001** |
| 365-day case fatality | 103,089 (55.0%) [54.7–55.2] | 1,421 (37.3%) [35.7–38.8] | 870 (38.5%) [36.5–40.5] | 2,509 (47.8 [46.5–49.2]) | 98,289 (55.8%) [55.5–56.0] | **−0.38** | <**0.001** | **−0.35** | <**0.001** | −0.16 | <**0.001** |

†Standardized difference and p-value between patients with schizophrenia and controls.

‡Standardized difference and p-value between patients with bipolar disorder and controls.

ƒStandardized difference and p-value between patients with major depressive disorder and controls.

SD ≤ |0.20| was chosen to indicate a negligible difference in the mean or prevalence of a variable between groups. SD > |0.20| shown in bold. P value < 0.05 shown in bold.

SMI, severe mental illness; 95% CI: 95% confidence interval.

**Table 2. Case fatality in septic shock patients with versus without SMI (1:up to 4 patients matched, within hospital, for age (5-year range), sex, degree of social deprivation, and year of hospitalization).**

| | Patients with schizophrenia | Matched controls | SD† | p-value† | Patients with bipolar disorder | Matched controls | SD‡ | p-value‡ | Patients with major depressive disorder | Matched controls | SDƒ | p-valueƒ |
|---|---|---|---|---|---|---|---|---|---|---|---|---|
| N | 3,269 | 10,894 | | | 1,923 | 6,303 | | | 4,432 | 14,452 | | |
| Primary outcome | | | | | | | | | | | | |
| 90-day case fatality–n (weighted %) [95% CI] | 1,052 (32.2%) [30.6–33.8] | 5,000 (45.5%) [43.7–47.2] | **−0.28** | **<0.001** | 632 (32.9%) [30.7–34.9] | 2,854 (45.3%) [43.0–47.5] | **−0.26** | **<0.001** | 1,834 (41.4%) [39.9–42.8] | 6,798 (47.1%) [45.6–48.5] | −0.11 | **<0.001** |
| Secondary outcomes | | | | | | | | | | | | |
| 30-day case fatality–n (weighted %) [95% CI] | 803 (24.6%) [23.0–26.0] | 4,092 (37.2%) [35.6–38.9] | **−0.28** | **<0.001** | 484 (25.2%) [23.2–27.2] | 2,375 (37.7%) [35.5–39.9] | **−0.27** | **<0.001** | 1,445 (32.6%) [31.2–34.0] | 5,604 (39.2%) [37.7–40.1] | −0.14 | **<0.001** |
| 365-day case fatality–n (weighted %) [95% CI] | 1,244 (38.1%) [36.4–39.7] | 5,675 (51.4%) [49.7–53.1] | **−0.27** | **<0.001** | 761 (39.6%) [37.8–41.8] | 3,232 (51.1%) [48.8–53.3] | **−0.23** | **<0.001** | 2,156 (48.7%) [47.2–50.1] | 7,678 (53.0%) [51.5–54.4] | −0.09 | **<0.001** |

*1:up to 4 patients matched, within a hospital, for age (5-year range), sex, degree of social deprivation, and year of hospitalization.

†Standardized difference and *p*-value between patients with schizophrenia and matched controls

‡Standardized difference and *p*-value between patients with bipolar disorder and matched controls.

ƒStandardized difference and *p*-value between patients with major depressive disorder and matched controls.

SD ≤ |0.20| was chosen to indicate a negligible difference in the mean or prevalence of a variable between groups. SD > |0.20| shown in bold. *P* value < 0.05 shown in bold.

SMI, severe mental illness; 95% CI, 95% confidence interval.

## Comparison of clinical profiles in septic shock patients with and without SMI

Patients with a major depressive disorder were more likely to have a tobacco (SD = 0.23) and alcohol (SD = 0.32) addiction, and patients with bipolar disorders were more likely to have an addiction to other substance than were their matched controls (SD = 0.22) (**Table 4**). Patients with schizophrenia and those with bipolar disorder had lower Charlson comorbidity index scores (SD = −0.27 and SD = −0.23, respectively), especially fewer malignancies (SD = −0.32 and SD = −0.26, respectively). Patients with bipolar disorder were more likely to have neurological failure than were their matched controls (SD = 0.25) (**S1 Table**). Differences in the site of infection or type of pathogen were negligible between SMI patients and their matched controls (**S2 Table**).

## Discussion

In this nationwide, population-based cohort study, the 30-, 90-, and 365-day CFRs in patients with septic shock admitted to the ICU were lower in patients with SMI than in other patients, after controlling for multiple potential confounding factors (using intrahospital matching and adjustments for multiple comorbidities and illness severity) and addressing potential biases not considered in previous studies [14–17].

The reasons for the differences in survival between patients with SMI and controls could not be determined in our study but may include differences in immunological profiles [35–39]

**Table 3. aHRs for 90-day case fatality in septic shock patients with SMI compared to those without (1:up to 4 patients matched, within hospital, for age (5-year range), sex, degree of social deprivation, and year of hospitalization).**

| | HR [95% CI] | p-value | HR [95% CI] | p-value | HR [95% CI] | p-value |
|---|---|---|---|---|---|---|
| Patients with schizophrenia (vs. matched controls) | 0.70 [0.65–0.75] | <**0.001** | - | - | - | - |
| Patients with bipolar disorder (vs. matched controls) | - | - | 0.70 [0.63–0.76] | <**0.001** | - | - |
| Patients with major depressive disorder (vs. matched controls) | - | - | - | - | 0.85 [0.81–0.90] | <**0.001** |
| Smoking addiction (yes vs. no) | 0.92 [0.84–1.00] | **0.049** | 0.90 [0.79–1.01] | 0.080 | 0.83 [0.77–0.90] | <**0.001** |
| Alcohol addiction (yes vs. no) | 0.94 [0.86–1.02] | 0.155 | 0.90 [0.79–1.02] | 0.091 | 0.89 [0.822–0.96] | **0.002** |
| Other substance addiction (yes vs. no) | 0.77 [0.63–0.95] | **0.014** | 0.74 [0.53–1.02] | 0.065 | 0.57 [0.46–0.71] | <**0.001** |
| Overweight or obese (yes vs. no) | 0.81 [0.74–0.88] | <**0.001** | 0.77 [0.70–0.86] | <**0.001** | 0.82 [0.77–0.88] | <**0.001** |
| Charlson index | | | | | | |
| 0 | 1.00 | - | 1.00 | - | 1.00 | - |
| 1–2 | 0.92 [0.84–1.01] | 0.095 | 1.15 [1.00–1.30] | <**0.001** | 1.06 [0.97–1.16] | 0.180 |
| ≥3 | 1.22 [1.11–1.33] | <**0.001** | 1.43 [1.26–1.62] | <**0.001** | 1.39 [1.28–1.50] | <**0.001** |
| Trauma (yes vs. no) | 0.54 [0.41–0.72] | <**0.001** | 0.61 [0.41–0.91] | **0.016** | 0.54 [0.40–0.72] | <**0.001** |
| Surgery (yes vs. no) | 0.75 [0.69–0.82] | <**0.001** | 0.95 [0.85–1.06] | 0.362 | 0.77 [0.72–0.83] | <**0.001** |
| SAPS II score at ICU admission | 1.03 [1.03–1.03] | <**0.001** | 1.03 [1.03–1.03] | <**0.001** | 1.03 [1.03–1.03] | <**0.001** |
| Respiratory failure (yes vs. no) | 1.04 [0.97–1.12] | 0.242 | 01.06 [0.97–1.16] | 0.225 | 1.10 [1.04–1.17] | <**0.001** |
| Renal failure (yes vs. no) | 0.81 [0.76–0.87] | <**0.001** | 0.79 [0.72–0.87] | <**0.001** | 0.83 [0.78–0.88] | <**0.001** |
| Neurologic failure (yes vs. no) | 1.01 [0.94–1.09] | 0.710 | 0.99 [0.90–1.09] | 0.871 | 0.94 [0.88–0.99] | **0.030** |
| Cardiovascular failure (yes vs. no) | 0.67 [0.61–0.74] | <**0.001** | 0.74 [0.65–0.83] | <**0.001** | 0.74 [0.69–0.80] | <**0.001** |
| Hematologic failure (yes vs. no) | 0.86 [0.79–0.94] | <**0.001** | 0.89 [0.79–1.00] | **0.048** | 0.94 [0.87–1.01] | 0.081 |
| Metabolic failure (yes vs. no) | 1.13 [1.11–1.33] | <**0.001** | 1.02 [0.92–1.13] | 0.751 | 1.14 [1.07–1.21] | <**0.001** |
| Hepatic failure (yes vs. no) | 1.74 [1.60–1.90] | <**0.001** | 1.71 [1.51–1.94] | <**0.001** | 1.65 [1.53–1.79] | <**0.001** |
| Source of hospital admission (home vs. transfer) | 0.96 [0.82–1.12] | 0.596 | 0.95 [0.78–1.16] | 0.629 | 0.88 [0.82–0.96] | **0.002** |
| Time to ICU admission (≤1 day vs. >1 day) | 0.74 [0.69–0.80] | <**0.001** | 0.80 [0.73–0.88] | <**0.001** | 0.74 [0.70–0.79] | <**0.001** |

aHR, adjusted hazard ratio; HR, hazard ratio; ICU, intensive care unit; SAPS II, Simplified Acute Physiology Score II; SMI, severe mental illness; 95% CI, 95% confidence interval.

P value < 0.05 shown in bold.

The adjusted model included SMI with additional covariates of smoking, alcohol, and other substance addiction (yes vs. no), overweight or obesity (yes vs. no), the Charlson comorbidity index (0, 1–2, ≥3), presence of trauma (yes vs. no), surgical intervention (yes vs. no), SAPS II score (modified, without age), organ failures (yes vs. no for each of respiratory, renal, neurologic, cardiovascular, hematologic, metabolic, hepatic), the source of hospital admission (home, transfer from other hospital ward), and time to ICU admission (≤1 vs. > 1 day).

and exposures to the immunomodulatory effects of psychotropic medications [40]. Immunological characteristics of patients with SMI have been reported for many years, related to effects of the psychiatric disease and the psychotropic treatments. All 3 SMI conditions are associated with dysregulated cytokine responses that may be protective in septic shock [41], as already suggested in autoimmune diseases such as multiple sclerosis [42], rheumatoid arthritis, and Crohn's disease [40]. Overexpression of specific pro-inflammatory cytokines such as interleukin (IL)-12 and interferon-gamma (IFN-γ) has been reported in SMI, as in autoimmune diseases, and may offset the immunosuppressive state induced by sepsis [40,41]. This finding may in part be related to the treatments received by patients with SMI, with psychotropic drugs including antidepressants [43–45], lithium [46], and antipsychotics [47,48] able to modulate the inflammatory response [35]. This hypothesis has been reinforced during the Coronavirus Disease 2019 (COVID-19) pandemic, during which fluoxetine [49] (an antidepressant) and chlorpromazine [50] (an antipsychotic) were suggested to have beneficial effects. Specifically, a Severe Acute Respiratory Syndrome Coronavirus 2 (SARS-CoV-2) animal model showed

**Table 4. Clinical profiles of septic shock patients with SMI compared to those without (1:up to 4 patients matched, within hospital, for age (5-year range), sex, degree of social deprivation, and year of hospitalization).**

| | Patients with schizophrenia | Matched controls | SD† | p-value† | Patients with bipolar disorder | Matched controls | SD‡ | p-value‡ | Patients with major depressive disorder | Matched controls | SDƒ | p-value |
|---|---|---|---|---|---|---|---|---|---|---|---|---|
| N | 3,269 | 10,894 | | | 1,923 | 6,303 | | | 4,432 | 14,452 | | |
| Age–year | | | | | | | | | | | | |
| Mean ± SD [95% CI] | 59.6 ± 13.5 [59.2–60.1] | 59.9 ± 7.4 [59.6–60.1] | 0.02 | 0.495 | 63.5 ± 12.3 [62.9–64.1] | 63.7 ± 6.8 [63.4–64.0] | −0.02 | 0.629 | 64.7 ± 7.3 [64.1–64.9] | 64.5 ± 13.4 [64.4–64.9] | −0.02 | 0.552 |
| Distribution–n (weighted %) [95% CI] | | | | 0.918 | | | | 0.872 | | | | 0.895 |
| 18–44 | 425 (13.0%) [11.8–14.2]) | 1,141 (12.6%) [11.5–13.8] | 0.01 | | 137 (7.1%) [6.0–8.2] | 364 (6.8%) [5.6–7.9] | 0.01 | | 307 (6.9%) [6.2–7.7] | 778 (6.6%) [5.9–7.3] | 0.01 | |
| 45–64 | 1,644 (50.3%) [48.8–52.0] | 5,504 (50.1%) [48.4–51.8] | 0.00 | | 818 (42.5%) [40.3–44.7] | 2,610 (42.3%) [40.0–44.5] | 0.01 | | 1,867 (42.1%) [40.7–43.6] | 5,967 (41.9%) [40.4–43.3]] | 0.01 | |
| 65–75 | 795 (24.3%) [22.8–25.7] | 2,794 (24.4) [22.8–25.8] | −0.00 | | 646 (33.6%) [31.4–35.7] | 2,189 (33.3%) [31.1–35.4] | 0.01 | | 1,233 (27.8%) [26.5–29.1] | 4,255 (28.2%) [26.5–29.1] | −0.01 | |
| >75 | 405 (12.4%) [11.3–13.5] | 1,455 (12.9) [11.7–14.0] | −0.02 | | 322 (16.7%) [15.1–18.4] | 1,140 (17.7%) [16.0–19.4] | −0.02 | | 1,025 (23.1%) [21.9–24.4] | 3,452 (23.3%) [21.9–24.4] | −0.00 | |
| Age at death–year | | | | | | | | | | | | |
| Mean ± SD [95% CI] | 63.1 ± 13.1 [62.3–63.9] | 62.6 ± 12.4 [61.7–62.4] | 0.04 | **0.044** | 67.0 ± 12.7 [66.1–67.9] | 66.1 ± 11.3 [65.3–66.2] | 0.08 | **0.045** | 67.1 ± 12.7 [66.5–67.7] | 67.4 ± 12.4 [66.6–67.2] | −0.02 | 0.710 |
| Sex (women)–n (weighted %) [95% CI] | 1,186 (36.3%) [34.6–37.9] | 3,740 (36.3%) [34.6–37.9] | 0.00 | 1.000 | 1,003 (52.2%) [50.0–54.4] | 3,115 (52.2%) [50.0–54.4] | 0.00 | 1.000 | 2,292 (51.7%) [50.2–53.2] | 7,144 (51.7%) [50.2–53.2] | 0.00 | 1.000 |
| Social deprivation, – n (weighted %) [95% CI] | | | | 1.000 | | | | 1.000 | | | | 1.000 |
| Least deprived | 1,067 (32.6%) [31.0–34.2] | 3,789 (32.6%) [31.0–34.2] | 0.00 | | 623 (32.4%) [30.3–34.5] | 2,231 (32.4%) [30.3–34.5] | 0.00 | | 1,278 (28.8%) [27.5–30.2] | 4,547 (28.8%) [27.5–30.2] | 0.00 | |
| Less deprived | 501 (15.3%) [14.1–16.6] | 1,670 (15.3%) [14.1–16.6] | 0.00 | | 290 (15.1%) [13.5–16.7] | 891 (15.1%) [13.5–16.7] | 0.00 | | 661 (14.9%) [13.9–16.0] | 2,062 (14.9%) [13.9–16.0] | 0.00 | |
| More deprived | 990 (30.3%) [28.7–31.9] | 3,178 (30.3%) [28.7–31.9] | 0.00 | | 629 (32.7%) [30.6–34.8] | 1,990 (32.7%) [30.6–34.8] | 0.00 | | 1,443 (32.6%) [31.2–33.9] | 4,581 (32.6%) [31.2–33.9] | 0.00 | |
| Most deprived | 711 (21.8%) [20.3–23.2] | 2,281 (21.8%) [20.3–23.2] | 0.00 | | 381 (19.8%) [18.0–21.6] | 1,191 (19.8%) [18.0–21.6] | 0.00 | | 1,050 (23.7%) [22.4–24.9] | 3,262 (23.7%) [22.4–24.9] | 0.00 | |
| Year–n (weighted %) [95% CI] | | | | 1.000 | | | | 1.000 | | | | 1.000 |
| 2014 | 571 (17.5%) [16.2–18.8] | 1,844 (17.5%) [16.2–18.8] | 0.00 | | 358 (18.6%) [16.9–20.4] | 1,157 (18.6%) [16.9–20.4] | 0.00 | | 822 (18.6%) [17.4–19.7] | 2,602 (18.6%) [17.4–19.7] | 0.00 | |
| 2015 | 659 (20.2%) [18.8–21.5] | 2,184 (20.2%) [18.8–21.5] | 0.00 | | 387 (20.1%) [18.3–21.9] | 1,243 (20.1%) [18.3–21.9] | 0.00 | | 908 (20.5%) [19.3–21.7] | 2,964 (20.5%) [19.3–21.7] | 0.00 | |
| 2016 | 669 (20.5%) [19.1–21.8] | 2,264 (20.5%) [19.1–21.8] | 0.00 | | 412 (21.4%) [19.6–23.3] | 1,357 (21.4%) [19.6–23.3] | 0.00 | | 915 (20.7%) [19.5–21.8] | 3,006 (20.7%) [19.5–21.8] | 0.00 | |

(*Continued*)

**Table 4.** (Continued)

| | Patients with schizophrenia | Matched controls | SD† | p-value† | Patients with bipolar disorder | Matched controls | SD‡ | p-value‡ | Patients with major depressive disorder | Matched controls | SDƒ | p-value |
|---|---|---|---|---|---|---|---|---|---|---|---|---|
| 2017 | 678 (20.7%) [19.1–22.1] | 2,321 (20.7%) [19.1–22.1] | 0.00 | | 371 (19.3%) [17.5–21.1] | 1,256 (19.3%) [17.5–21.1] | 0.00 | | 889 (20.1%) [18.9–21.2] | 2,925 (20.1%) [18.9–21.2] | 0.00 | |
| 2018 | 692 (21.2%) [19.7–22.6] | 2,281 (21.2%) [19.7–22.6] | 0.00 | | 395 (20.5%) [18.7–22.3] | 1,290 (20.5%) [18.7–22.3] | 0.00 | | 898 (20.3%) [19.1–21.5] | 2,955 (20.3%) [19.1–21.5] | 0.00 | |
| Smoking addiction–n (weighted %) [95% CI] | 766 (23.4%) [22.0–24.9] | 2,133 (19.6) [18.3–21.0] | 0.09 | <**0.001** | 422 (21.9%) [20.1–23.8] | 1,098 (17.3%) [15.6–19.0]) | 0.12 | <**0.001** | 1,199 (27.1%) [25.7–28.4] | 2,552 (17.7) [16.6–18.9] | **0.23** | <**0.001** |
| Alcohol addiction–n (weighted %) [95% CI] | 600 (18.4%) [17.0–19.7] | 2,136 (19.8%) [18.5–21.2] | −0.04 | 0.129 | 445 (23.1%) [21.2–25.0] | 1,011 (16.2%) [14.6–17.9] | 0.17 | <**0.001** | 1,261 (28.5%) [27.1–29.8] | 2,225 (15.5%) [14.4–16.5] | **0.32** | <**0.001** |
| Other substance addiction–n (weighted %) [95% CI] | 227 (6.9%) [6.1–7.8] | 311 (2.9%) [2.3–3.4] | 0.19 | <**0.001** | 115 (6.0%) [4.9–7.0] | 110 (1.7%) [1.2–2.3] | **0.22** | <**0.001** | 220 (5.0%) [4.2–5.6] | 234 (1.6%) [1.3–2.0] | 0.19 | <**0.001** |
| Opioid-related Disorder | 103 (3.2%) [2.6–3.7] | 155 (1.6%) [1.2–2.0] | 0.10 | <**0.001** | 41 (2.1%) [1.5–2.8] | 54 (0.9%) [0.5–1.4] | 0.10 | **0.004** | 110 (2.5%) [2.0–3.0] | 133 (0.6%) [0.3–0.8] | 0.12 | <**0.001** |
| Cannabis-related Disorder | 79 (2.4%) [1.9–2.9] | 69 (0.7%) [0.4–1.0] | 0.14 | <**0.001** | 34 (1.8%) [1.2–2.4] | 31 (0.5%) [0.2–0.8] | 0.12 | <**0.001** | 44 (1.0%) [0.7–1.2] | 66 (0.5%) [0.3–0.7] | 0.06 | **0.004** |
| Cocaine-related disorder | 32 (1.0%) [0.6–1.3] | 33 (0.4%) [0.2–0.6] | 0.08 | **0.003** | 15 (0.8%) [0.4–1.2] | 8 (0.1%) [0.0–0.3] | 0.10 | **0.008** | 22 (0.5%) [0.3–0.7] | 30 (0.2%) [0.06–0.3] | 0.05 | **0.019** |
| Other substances | 115 (3.5%) [2.9–4.1] | 100 (1.0%) [0.06–1.3%] | 0.17 | <**0.001** | 57 (3.0%) [2.2–3.7] | 42 (0.6%) [0.3–1.0] | 0.18 | <**0.001** | 111 (2.5%) [2.0–3.0] | 81 (0.6%) [0.3–0.8] | 0.16 | <**0.001** |
| Overweight or obese–n (weighted %) [95% CI] | 533 (16.3%) [15.0–17.6] | 1,945 (17.7%) [16.3–19.0] | −0.04 | 0.148 | 411 (21.4%) [19.5–23.2] | 1,194 (18.9%) [17.1–20.6] | 0.06 | 0.053 | 1,019 (23.0%) [21.7–24.2] | 2,911 (20.5%) [19.4–21.7]) | 0.06 | **0.005** |
| Charlson index–n (weighted %) [95% CI] | | | | <**0.001** | | | | <**0.001** | | | | <**0.001** |
| 0 | 1,104 (33.8%) [32.2–35.4] | 2,223 (21.4 [20.0–22.8]) | **0.28** | | 567 (29.5%) [27.4–31.5] | 1,186 (20.1%) [18.3–21.9] | **0.22** | | 812 (18.3%) [17.2–19.5] | 2,762 (19.8%) [17.1–19.5] | −0.04 | |
| 1–2 | 1,036 (31.7%) [30.1–33.3] | 3,312 (30.7 [29.1–32.3]) | 0.02 | | 621 (32.3%) [30.2–34.4] | 1,947 (30.5%) [28.4–32.5] | 0.04 | | 1,255 (28.3%) [26.9–29.6] | 4,471 (31.3%) [26.9–29.6] | −0.07 | |
| ≥3 | 1,129 (34.5%) [32.9–36.2] | 5,359 (47.9%) [46.2–49.6] | **−0.27** | | 735 (38.2%) [36.0–40.4] | 3,170 (49.4%) [47.2–51.7] | **−0.23** | | 2,365 (53.4%) [51.9–54.8] | 7,219 (48.9%) [47.5–50.4] | 0.09 | |
| Trauma–n (weighted %) [95% CI] | 98 (3.0%) [2.4–3.6] | 252 (2.3) [1.8–2.8] | 0.04 | 0.090 | 36 (1.9%) [1.2–2.5] | 122 (1.8%) [1.2–2.3] | 0.01 | 0.817 | 55 (1.2%) [0.9–1.5] | 224 (1.5%) [1.2–1.9] | −0.02 | 0.259 |
| Surgery–n (weighted %) [95% CI] | 594 (18.2%) [16.8–19.5] | 2,459 (22.1%) [20.6–23.5] | −0.10 | <**0.001** | 351 (18.3%) [16.5–20.0] | 1,511 (23.8% [21.9–25.7] | −0.14 | <**0.001** | 935 (21.1%) [19.9–22.3] | 3,397 (23.4%) [22.1–24.6] | −0.06 | <**0.001** |
| SAPS II score at ICU admission, Mean ± SD [95% CI] | 42.8 ± 21.7 [42.0–43.5] | 44.8 ± 12.5 [44.4–45.2] | −0.11 | <**0.001** | 43.3 ± 22.7 [42.3–44.3] | 44.0 ± 12.7 [43.4–44.5] | −0.04 | 0.347 | 43.2 ± 22.4 [42.5–43.9] | 43.8 ± 12.8 [43.4–44.2] | −0.03 | 0.210 |

*(Continued)*

**Table 4.** (Continued)

| | Patients with schizophrenia | Matched controls | SD† | p-value† | Patients with bipolar disorder | Matched controls | SD‡ | p-value‡ | Patients with major depressive disorder | Matched controls | SDƒ | p-value |
|---|---|---|---|---|---|---|---|---|---|---|---|---|
| **Site of infection–n (weighted %) [95% CI]** | | | | | | | | | | | | |
| Respiratory | 1,568 (48.0%) [46.3–49.7] | 4,537 (41.5%) [39.8–43.2] | 0.13 | **<0.001** | 826 (43.0%) [40.7–45.2] | 2,486 (38.8%) [36.6–41.0] | 0.09 | **0.009** | 1,888 (42.6%) [41.1–44.0] | 5,719 (38.9%) [37.4–40.3] | 0.08 | **<0.001** |
| Gastrointestinal | 521 (15.9%) [14.7–17.2] | 1,714 (16.0%) [14.7–17.3] | −0.00 | 0.944 | 290 (15.1%) [13.5–16.7] | 1,166 (18.7%) [17.0–20.4] | −0.10 | **0.003** | 704 (15.9%) [14.8–16.9] | 2,533 (17.6%) [16.4–18.7] | −0.05 | **0.029** |
| Renal | 311 (9.5%) [8.5–10.5] | 894 (7.9%) [6.9–8.8] | 0.06 | **0.019** | 216 (11.2%) [9.8–12.6] | 533 (8.3%) [7.0–9.5] | 0.10 | **0.002** | 472 (10.7%) [9.7–11.6] | 1,318 (9.2%) [8.3–10.1] | 0.05 | **0.026** |
| Cardiac | 306 (9.4%) [8.4–10.4] | 1,138 (10.4%) [9.3–11.4] | −0.03 | 0.165 | 161 (8.4%) [7.1–9.6] | 701 (10.9%) [9.4–12.2] | −0.09 | **0.008** | 456 (10.3%) [9.4–11.2] | 1,535 (10.4%) [9.5–11.3] | −0.00 | 0.862 |
| Dermatologic | 180 (5.5%) [4.7–6.3] | 783 (6.9%) [6.1–7.8] | −0.06 | **0.017** | 96 (5.0%) [4.0–6.0] | 427 (6.6%) [5.5–7.6] | −0.07 | **0.038** | 271 (6.1%) [5.4–6.8] | 1,008 (7.0%) [6.2–7.7] | −0.03 | 0.110 |
| **Organ failures–n (weighted %) [95% CI]** | | | | | | | | | | | | |
| Respiratory | 2,069 (63.3%) [61.6–64.9] | 6,440 (58.9%) [57.2–60.6] | 0.09 | **<0.001** | 1,197 (62.3%) [60.0–64.4] | 3,659 (58.7%) [56.5–60.9] | 0.07 | **0.024** | 2,708 (61.1%) [59.7–62.5] | 8,368 (57.8%) [56.3–59.3] | 0.07 | **0.002** |
| Renal | 1,286 (39.3%) [37.7–41.0] | 5,373 (48.2%) [46.5–49.9] | −0.18 | **<0.001** | 815 (42.4%) [40.2–44.6] | 3,145 (49.3%) [47.0–51.5] | −0.14 | **<0.001** | 1,993 (45.0%) [43.5–46.4] | 7,277 (49.9%) [48.4–51.4] | −0.10 | **<0.001** |
| Neurologic | 1,051 (32.2%) [30.5–33.8] | 2,575 (23.7%) [22.2–25.1] | 0.19 | **<0.001** | 659 (34.3%) [32.1–36.4] | 1,473 (23.1%) [21.2–25.0] | **0.25** | **<0.001** | 1,288 (29.1%) [27.7–30.4] | 3,393 (23.3%) [22.0–24.5] | 0.13 | **<0.001** |
| Cardiovascular | 428 (13.1%) [11.9–14.2] | 1,577 (13.9%) [12.7–15.1] | −0.02 | 0.359 | 239 (12.4%) [10.9–13.9] | 1,036 (15.9%) [14.3–17.6] | −0.10 | **0.002** | 660 (14.9%) [13.8–15.9] | 2,291 (15.8%) [14.7–16.8] | −0.02 | 0.252 |
| Hematologic | 395 (12.1%) [11.0–13.2] | 1,705 (15.9%) [14.7–17.2] | −0.11 | **<0.001** | 212 (11.0%) [9.6–12.4] | 960 (15.2%) [13.6–16.8] | −0.12 | **<0.001** | 602 (13.6%) [12.6–14.6] | 2,086 (14.5%) [13.5–15.5] | −0.03 | 0.212 |
| Metabolic | 673 (20.6%) [19.2–22.0]) | 2,554 (23.1%) [21.7–24.6] | −0.06 | **0.013** | 404 (21.0%) [19.2–22.8] | 1,468 (23.5%) [21.6–25.3] | −0.06 | 0.068 | 1,018 (23.0%) [21.7–24.2] | 3,448 (23.5%) [22.2–24.7] | −0.01 | 0.553 |
| Hepatic | 237 (7.3%) [6.3–8.2] | 1,439 (12.9%) [11.7–14.0] | −0.19 | **<0.001** | 145 (7.5%) [6.4–8.7] | 725 (11.5%) [10.0–12.9] | −0.14 | **<0.001** | 476 (10.7%) [9.8–11.7] | 1,676 (11.6%) [10.6–12.5] | −0.03 | 0.213 |
| **ICU supportive therapies–n (weighted %) [95% CI]** | | | | | | | | | | | | |
| Cardiopulmonary resuscitation | 161 (4.9%) [4.2–5.7] | 669 (6.2%) [5.3–7.0] | −0.05 | 0.031 | 85 (4.4%) [3.5–5.4] | 349 (5.6%) [4.6–6.6] | −0.05 | 0.096 | 177 (4.0%) [3.42–4.6] | 829 (5.6%) [4.9–6.2] | −0.07 | **<0.001** |
| Invasive mechanical ventilation | 2,787 (85.3%) [84.0–86.5] | 8,960 (82.0%) [80.7–83.3] | 0.09 | **<0.001** | 1,602 (83.3%) [81.6–85.0] | 5,076 (80.5%) [78.7–82.2] | 0.07 | 0.023 | 3,564 (80.4%) [79.8–81.5] | 11,556 (79.8%) [78.6–80.9] | 0.02 | 0.459 |

(*Continued*)

**Table 4.** (Continued)

| | Patients with schizophrenia | Matched controls | SD† | p-value† | Patients with bipolar disorder | Matched controls | SD‡ | p-value‡ | Patients with major depressive disorder | Matched controls | SDƒ | p-value |
|---|---|---|---|---|---|---|---|---|---|---|---|---|
| Renal replacement therapy | 672 (20.6%) [19.2–21.9] | 3,278 (30.0%) [28.0–31.1] | −0.21 | **<0.001** | 452 (23.5%) [21.6–25.4] | 1,914 (30.0%) [27.9–32.0] | −0.15 | **<0.001** | 1,060 (23.9%) [22.7–25.2] | 4,149 (28.8%) [27.4–30.1]) | −0.11 | **<0.001** |
| Transfusion | 969 (29.6%) [28.1–31.2] | 3,782 (34.6%) [33.0–36.3] | −0.11 | **<0.001** | 540 (28.1%) [26.1–30.1] | 2,228 (34.9%) [32.8–37.1] | −0.15 | **<0.001** | 1,475 (33.3%) [31.9–34.7] | 4,940 (33.8%) [32.4–35.2] | −0.01 | 0.578 |
| Source of hospital admission–n (weighted %) [95% CI] | | | | | | | | | | | | |
| Home | 3,040 (93.0%) [92.1–93.9] | 10,577 (97.1%) [96.5–97.6] | −0.19 | **<0.001** | 1,814 (94.3%) [93.3–95.4] | 6,075 (96.2%) [95.3–97.1] | −0.09 | **0.006** | 4,212 (95.0%) [94.4–95.7] | 13,952 (96.4%) [95.8–96.9] | −0.07 | **0.002** |
| Transfer from other hospital | 229 (7.0%) [6.1–7.9] | 317 (2.9%) [2.4–3.5] | 0.19 | | 109 (5.7%) [4.6–6.7] | 228 (3.8%) [2.9–4.6] | 0.09 | | 220 (5.0%) [4.3–5.6] | 500 (3.6%) [3.0–4.1] | 0.07 | |
| Time to ICU admission ≤1 day–n (weighted %) [95% CI] | 2,180 (67.0%) [65.1–68.3] | 6,734 (62.2%) [60.5–63.8] | 0.09 | **<0.001** | 1,298 (67.5%) [65.4–69.6] | 3,848 (61.2%) [59.0–63.4] | 0.13 | **<0.001** | 2,721 (61.4%) [60.0–62.8] | 8,903 (61.2%) [59.7–62.6] | 0.00 | 0.823 |
| Hospital characteristics–n (weighted %) [95% CI] | | | | 1.000 | | | | 1.000 | | | | 1.000 |
| Academic | 1,532 (46.9%) [45.2–48.6] | 5,777 (46.86 [45.2–48.6]) | 0.00 | | 840 (43.7%) [41.5–45.9] | 3,187 (43.7%) [41.5–45.9] | 0.00 | | 1,871 (42.2%) [40.8–43.7] | 7,051 (42.2%) [40.8–43.7] | 0.00 | |
| Other public hospital | 1,637 (50.1%) [48.4–51.8] | 4,898 (50.1%) [48.4–51.8] | 0.00 | | 1,006 (52.3%) [50.1–54.5] | 2,947 (52.3%) [50.1–54.5] | 0.00 | | 2,391 (54.0%) [52.5–55.4] | 7,011 (54.0%) [52.5–55.4] | 0.00 | |
| Private | 100 (3.1%) [2.5–3.7] | 219 (3.1%) [2.5–3.7] | 0.00 | | 77 (4.0%) [3.1–4.8] | 169 (4.0%) [3.1–4.8] | 0.00 | | 170 (3.8%) [3.3–4.4] | 390 (3.8%) [3.3–4.4] | 0.00 | |

†Standardized difference and p-value between patients with schizophrenia and matched controls.

‡Standardized difference and p-value between patients with bipolar disorder and matched controls.

ƒStandardized difference and p-value between patients with major depressive disorder and matched controls.

SD ≤ |0.20| was chosen to indicate a negligible difference in the mean or prevalence of a variable between groups. SD > |0.20| shown in bold. P value < 0.05 shown in bold.

ICU, intensive care unit; SMI, severe mental illness; 95% CI: 95% confidence interval.

independent antiviral and anti-inflammatory effects of fluoxetine [51], in line with several observational studies [44,52,53]. A candidate mechanism, shared by several psychotropic medications and supported by several preclinical [54] and observational studies [44,52,55,56], is the functional inhibition of acid sphingomyelinase (FIASMA) leading to a regulation of apoptosis, cellular differentiation, proliferation, and cell migration. Finally, several RCTs and observational studies have reported evidence of efficacy of fluvoxamine at a daily dose of 200 mg or more against COVID-19 among outpatients with COVID-19 [57–59] and COVID-19 ICU patients [60]. A potential implication of these results is that the frequently observed discontinuation of psychotropic medications on admission to the ICU should be carefully

considered given the risks of relapse of the psychiatric disorder as well as the potential benefits of these drugs on mortality in the context of septic shock. Further studies are needed to explore these immune and pharmacological mechanisms.

The long-term goal of identifying patient groups with higher case fatality in sepsis than that in the general population is to identify the mechanisms underlying the outcome differences and, critically, modifiable mechanisms that can serve as targets for interventional approaches geared to reduce the outcome disparity of the affected group in reference to the general population. A key finding of our study (and most of the prior ones [14–17]) is that some factors unique to patients with SMI (e.g., possibly baseline immune dysfunction leading to a different, more protective, response to infection) not only negated the adverse prognostic effects of SMI in septic shock patients (which could have resulted in similar case fatality between the groups), but were associated with markedly lower case fatality among these patients. The magnitude of this effect estimate is remarkable, especially in this vulnerable population marked by low socio-economic status. A major implication is that future work to characterize potential differences in response to infection among patients with and without SMI across key domains of the immune system may identify potential targets for therapeutic interventions to reduce short-term mortality in the general population. However, there were some important differences between patients with and without SMI after matching (e.g., fewer malignancies and fewer comorbid conditions), which may have influenced outcomes. Although these differences were adjusted for, it is possible that residual confounding remained. In addition, the social deprivation indicator is based on the area level and may thus also lead to residual confounding.

The lower CFR may have health policy implications on future focus of resource allocation to improve life expectancy in patients with SMI. This finding suggests that the higher mortality rate due to infection/sepsis among patients with SMI reported in previous studies [2] appears to be due to the increased risk of infection/sepsis among patients with SMI and potentially poorer access to timely and adequate care, but not due to greater case fatality once they have been hospitalized for septic shock. As a consequence, our findings suggest that effective primary prevention interventions (i.e., before the onset of infection, to reduce the incidence of infection in patients with SMI) should be prioritized. However, evidence-based strategies for the prevention of infection in patients with SMI are scarce, as highlighted by a recent review on the prevalence rates and immunogenicity of vaccinations in patients with SMI [61]. Future studies should confirm this hypothesis on the full sample of individuals with SMI and sepsis in the population.

Our study has several limitations. First, we described only patients who died in hospital, which means that the CFR might be underestimated. Deaths occurring outside the hospital are extremely rare in France but could be differentially experienced by people with SMI [28]. Nonetheless, our findings at 30 and 90 days were similar to those reported in other studies [62]. In addition, the evolution of the CFR between 30 and 90 days and between 90 and 365 days was similar in the patients with and without SMI, supporting a lack of bias to account for the different extrahospital mortality. Second, a weakness of administrative databases is the potential miscoding of diagnoses during hospital stays, which can underestimate important patient features (especially for overweight and obesity, which are insufficiently coded in administrative databases but which allow the most serious cases to be targeted for epidemiological research [63,64]) and disease severity at ICU admission. Missing data are thus assumed to indicate no disease present. In addition, the key exposure in the present study (i.e., SMI) can be misclassified due to use of ICD-10 codes, which could have affected reported effect estimates. Misclassification of mental disorders would be expected to blur the differences between groups and thus diminish outcome differences between septic shock patients with and without SMI. This would suggest that the study's findings may represent possible underestimation of

the magnitude of the better outcomes observed among patients with SMI. However, the coding has been strictly regulated since the DRG system was introduced in France. To control for these weaknesses, we used a matching procedure and adjustment based on a large number of patient characteristics and controlling for confounders at the hospital level. The matching process failed for 15% of patients due to the age imbalance between patients with and without SMI. However, the sensitivity analysis on the whole cohort reported similar findings. There are also limitations associated with the lack of some variables, including specific description of psychotropic medications, body mass index, fitness, and blood lactate levels, which could be useful to categorize our patients. Furthermore, the time between the onset of infection and the need for vasopressor support could not be determined. Some patients may require vasopressor support for a problem other than septic shock. Finally, processes of care for sepsis were not analyzed in detail in our study and may have differed across compared groups, which could have led to residual confounding in modelled effects. Patients with SMI are well documented to receive poorer quality of healthcare, in addition to stigma, stereotyping, and negative attitudes towards these patients by clinicians. Such care differences would be not be expected, however, to result in better outcomes of septic patients with SMI. Such potential differences in care processes would suggest that the study's findings may represent possible underestimation of the magnitude of the better outcomes observed among patients with SMI.

In conclusion, our findings suggest that SMI patients have a better outcome from septic shock in the ICU than those without SMI. This better prognosis may be explained by different immunological mechanisms and exposures to psychotropic medications. Further studies on these mechanisms that may potentially modulate outcomes may have important implications for all septic shock patients.

## Supporting information

**S1 STROBE Checklist. STROBE statement—Checklist of items that should be included in reports of *cohort studies*.**
(DOCX)

**S1 Fig. Forest plots of unadjusted (model 1) and adjusted hazard ratios (main model and sensitivity analyses) for 90-day hospital septic shock case fatality in septic shock patients with severe mental illness compared to those without (1:up to 4 patients matched, within hospital, for age (5-year range), sex, degree of social deprivation, and year of hospitalization).**
(DOCX)

**S2 Fig. Forest plots of unadjusted (model 1) and adjusted hazard ratios (main model and sensitivity analyses) for 30-day hospital septic shock case fatality between septic shock patients with versus without severe mental illness (1:up to 4 patients matched, within hospital, for age (5-year range), sex, degree of social deprivation, and year of hospitalization).**
(DOCX)

**S3 Fig. Forest plots of unadjusted (model 1) and adjusted hazard ratios (main model and sensitivity analyses) for 1-year septic shock case fatality between septic shock patients with versus without severe mental illnesses (1:up to 4 patients matched, within hospital, for age (5-year range), sex, degree of social deprivation, and year of hospitalization).**
(DOCX)

**S4 Fig. Kaplan–Meier estimates of overall survival at 1 year after intensive care unit (ICU) admission in septic shock patients with and without severe mental illness (1:up to 4**

**patients matched, within hospital, for age (5-year range), sex, degree of social deprivation, and year of hospitalization).** (**A**) Overall survival in septic shock patients with schizophrenia compared to matched controls without severe mental illness. (**B**) Overall survival in septic shock patients with bipolar disorder compared to matched controls without severe mental illness. (**C**) Overall survival in septic shock patients with major depressive disorder compared to matched controls without severe mental illness.
(DOCX)

**S1 Table. Charlson comorbidities of septic shock patients with and without severe mental illness**\*.
(DOCX)

**S2 Table. Pathogens in septic shock patients with and without severe mental illness**\*.
(DOCX)

## Author Contributions

**Conceptualization:** Marc Leone, Veronica Orleans, Laurent Boyer, Guillaume Fond.

**Data curation:** Veronica Orleans.

**Formal analysis:** Vanessa Pauly, Kossi Josue Srougbo, Sambou Diao, Sara Fernandes.

**Funding acquisition:** Laurent Boyer, Guillaume Fond.

**Investigation:** Laurent Boyer.

**Methodology:** Laurent Boyer, Guillaume Fond.

**Project administration:** Laurent Boyer, Guillaume Fond.

**Resources:** Laurent Boyer.

**Supervision:** Marc Leone, Jean-Louis Vincent, Laurent Boyer, Guillaume Fond.

**Validation:** Marc Leone, Pierre-Michel Llorca, Marco Solmi, Christoph U. Correll, Sara Fernandes, Jean-Louis Vincent, Laurent Boyer, Guillaume Fond.

**Writing – original draft:** Ines Lakbar.

**Writing – review & editing:** Marc Leone, Pierre-Michel Llorca, Marco Solmi, Christoph U. Correll, Sara Fernandes, Jean-Louis Vincent, Laurent Boyer, Guillaume Fond.

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
