## [Editor Report · Decision Letter 0]

19 Oct 2022

Dear Dr Boyer, 

Thank you for submitting your manuscript entitled "Severe mental illness is associated with a better prognosis in septic shock: results from 187,587 hospitalizations in France" for consideration by PLOS Medicine.

Your manuscript has now been evaluated by the PLOS Medicine editorial staff as well as by an academic editor with relevant expertise and I am writing to let you know that we would like to send your submission out for external peer review.

Please re-submit your manuscript within two working days, i.e. by Oct 21 2022 11:59PM.

Kind regards,

Philippa Dodd, MBBS MRCP PhD

Editor

PLOS Medicine

---

## [Decision Letter · Decision Letter 1]

8 Dec 2022

Dear Dr. Boyer,

Thank you very much for submitting your manuscript "Severe mental illness is associated with a better prognosis in septic shock: results from 187,587 hospitalizations in France" (PMEDICINE-D-22-03397R1) for consideration at PLOS Medicine. 

[LINK]

In light of these reviews, I am afraid that we will not be able to accept the manuscript for publication in the journal in its current form, but we would like to consider a revised version that addresses the reviewers' and editors' comments. Obviously we cannot make any decision about publication until we have seen the revised manuscript and your response, and we plan to seek re-review by one or more of the reviewers. 

We expect to receive your revised manuscript by Dec 29 2022 11:59PM. Please email us (plosmedicine@plos.org) if you have any questions or concerns.

We look forward to receiving your revised manuscript. 

Sincerely,

Philippa Dodd, MBBS MRCP PhD

PLOS Medicine

plosmedicine.org

GENERAL 

Please respond to all editor and reviewer comments detailed below, in full.

Please number the pages and lines in the manuscript starting at 1 on the abstract title and in continuous sequence thereafter.

Thank you for reporting your study according to STROBE. Please include the completed STROBE checklist as Supporting Information and refer to it in your reporting statement in the methods section. When completing the checklist, please use section and paragraph numbers, rather than page or line numbers.

Please consider how you frame your research question and the terminology you use when describing it – see comments below from the academic editor and reviewers.

The title, parts of the abstract, introduction and discussion imply that this is a study of whether people with serious mental illness, have lower mortality from sepsis. However, both the reviewers and the academic editor have raised concerns regarding this inference for multiple reasons, which the editorial team are in agreement with. Please see their detailed comments below and revise accordingly.

COMMENTS FROM THE ACADEMIC EDITOR

I agree with the decision about major revision, but I have the following additional comments.

Building on the comments from reviewer 1, there does seem to be a confusion around the research question being addressed and, as a consequence, the interpretation of the findings. I understand the research question to effectively be 'after adjusting for all the other factors (social, physical health, habits) that make people with SMI at risk of excess mortality, is there something about SMI as a disease process/its pharmacological treatment that might affect risk of mortality in those who have sepsis and who manage to access ICU ?'. This is not a study of whether people with SMI as a group have lower mortality from sepsis as implied by the title, the abstract conclusion and in other places in the paper. As the authors start to raise in the discussion, it is very likely that we are seeing a survival bias in the sample being examined - this is not the full sample of people with SMI and sepsis in the population. It does not include people with SMI and sepsis who had sepsis and died in the community, in the hospital before ICU access or, indeed, in the first 48 hours of ICU care. The actual research question being addressed has merit, but it should not be conflated with a research question about mortality from sepsis in people with SMI in general. For the background, I note that excess mortality in people with SMI is strongly related to social disadvantage - please add that to this list alongside unhealthy behaviours, increased infection risk, etc.

Please give further explanation for the exclusion of people with a diagnosis of sepsis in the first 48 hours of ICU admission.

Another reviewer raised the need to look in more detail into the potential reasons for the difference in mortality that might not be due to immunological profiles/medicines. There are some differences, not all significant but nonetheless present - lower malignancies and lower co-morbid conditions. Although these are adjusted for, it is possible that there is residual confounding. The SES indicators is also area level and may lead to residual confounding (people with SMI more likely to be from private hospitals indicates that there could actually be SES differences that are inadequately measured).

The level of 'other substance addiction' seems to be very high (in both people with SMI and those without). I might have missed it, but can you add the main substances that were contributing to this (if you have the information).

TITLE

Please revise your title in context of the academic editor and reviewer comments and according to PLOS Medicine's style. Your title must be nondeclarative and not a question. It should begin with the main concept if possible. "Effect of" should be used only if causality can be inferred, i.e., for an RCT. Please place the study design ("A randomized controlled trial," "A retrospective study," "A modelling study," etc.) in the subtitle (ie, after a colon).

DATA AVAILABILITY STATEMENT

Thank you for including a Data Availability Statement (DAS) which requires revision. For each data source used in your study: 

ABSTRACT

Abstract Background:

Please see academic editor comments above and reviewer comments below, which we agree with, and revise the abstract background accordingly. Provide the context of why the study is important. The final sentence should clearly state the study question.

Abstract Methods and Findings:

“90-day hospital mortality” – did all these patients die in hospital? If not which we suspect is the case then please report “90-day mortality”. See below (methods and results) also, please check and clarify or amend throughout as necessary.

Statistical reporting is a bit confusing here:

- It is unclear whether the results you present are adjusted or unadjusted (see statistical reviewer comments also) if you mention both would it be reasonable to report both?

- Detailing the number of controls within each group as you have done distracts from the data reporting and because this information is placed next to percentages it is easy to think these numbers contribute to deriving the percentages, but they do not. Please revise.

- Numerators and denominators used to derive percentages should be clearly reported 

- Where 95% CIs are reported, please also report p-values

- Suggest reporting statistical information as follows to improve accessibility to the reader: “HR 0.83, 95%CI [0.79, 0.88], p =/<” commas separating upper and lower confidence limits may help to mitigate against any confusion regarding negative values. Where p-values are significant please report as less than the significance level (<0.01 or <0.05 an so on)

In the last sentence of the Abstract Methods and Findings section, please describe the main limitation(s) of the study's methodology

Abstract conclusions:

Please see above and revise in-line with the reviewer and academic editor comments below.

Please address the study implications without overreaching what can be concluded from the data and interpret the study based on the results presented in the abstract, emphasizing what is new without overstating your conclusions.

Please avoid vague statements such as "these results have major implications for policy/clinical care". Mention only specific implications substantiated by the results.

Please avoid assertions of primacy ("We report for the first time....")

AUTHOR SUMMARY

INTRODUCTION

Please revise your introduction in mind of the academic editor and reviewer comments 

Please ensure you have addressed past research and explain the need for and potential importance of your study. Indicate whether your study is novel and how you determined that. If there has been a systematic review of the evidence related to your study (or you have conducted one), please refer to and reference that review and indicate whether it supports the need for your study. 

Please conclude the Introduction with a clear description of the study question or hypothesis.

METHODS and RESULTS

Under paragraph titled “outcomes” please elaborate on what you mean by “clinical characteristics”

Under paragraph titled “collected data” suggest “(…substance use)”

What does “origin of the patient” mean? Please clarify perhaps with the use of an alternative term

Where 95% CIs are reported, please also report p-values and the statistical tests used to determine them.

As in the abstract, suggest reporting statistical information as follows to improve accessibility to the reader: “HR 0.83, 95%CI [0.79, 0.88], p =/<” commas separating upper and lower confidence limits may help to mitigate against any confusion regarding negative values. Where p-values are significant please report as less than the significance level (<0.01 or <0.05 for example).

Under paragraph titled “comparison of mortality rates”: You state: “similar findings for 30-, 90- and 365-day hospital mortality” do you report hospital mortality or were some patients discharged? Please clarify and as necessary amend removing the word “hospital” so as to report “30-, 90-, 365-day mortality”. Please check carefully and revise throughout, including in the abstract.

TABLES

Please provide a table showing the baseline characteristics of the study population in the main manuscript as table 1.

Tables 1 and 2: Please quantify the results with p-values and 95% CI. If there is a specific reason (s) why it would be preferable to report standardized difference, please clearly state those reasons.

FIGURES 

Please see reviewer 2’s (statistical reviewer) comments regarding the presentation of the models

Figure 1: Please define ICU, we suggest in an expanded figure title

Figure 2: Thank you for including unadjusted analyses. Please remove the word “hospital” from the title. Please indicate the meaning of the dots and lines in the figure caption

Please address the same in the supporting figures

DISCUSSION

Please see reviewer 1 comments below regarding interpretation/implication of your study findings of your data.

Please expand the reporting of limitations in-line with reviewer comments

Please remove the sub-heading “conclusions” such that the discussion reads as a single piece of continuous prose. 

Please remove the conflict of interest statement from the main manuscript and place only in the submission form when you re-submit your manuscript.

SUPPORTING INFORMATION

See above (figures)

Supplementary figure 3: please define ICU in the caption/title

Supplementary tables 1, 3, 4: Please quantify the results with p-values and 95% CI. If there is a specific reason (s) why it would be preferable to report standardized difference, please clearly state those reasons.

Comments from the reviewers:

Reviewer #1: The manuscript describes the association of three categories of mental disorders with mortality among ICU-managed patients with septic shock in a population-based cohort.

The findings of this work extend to the French population those from recent population-based studies from Germany and the United States showing, unexpectedly, lower short-term mortality among hospitalized septic patients with mental disorders compared to those without these disorders.

The study design and the analytical approach are appropriate, and the study findings are well-described. The inclusion of severity of illness scores, overweight/obesity, and type of hospital, noted as key rationale for the present vs prior studies, in risk-adjustment models strengthens the internal validity of the reported prognostic associations of mental disorders in sepsis. Notably, however, severity of illness scores, overweight/obesity, and type of hospital did not differ statistically between the groups with vs without the examined mental disorders. The authors further extend prior reports on this topic by showing that the lower mortality in septic patients with mental disorders persists over short- , intermediate- and long-term periods.

Major comments

1. The authors state in the Discussion that their findings "are paradoxically in contrast with the excess mortality due to infection in patients with SMI from previous studies [2]." However, this statement represents an erroneous comparison of studies of two different types of death rates: a study showing low case fatality (the present study) and one with increased mortality rates (the cited study and similar types of studies). The two types of death rates and the corresponding descriptive terms are not interchangeable and an illustrative discussion of their use can be found in Laupland KB, et al. Chest 2021;159:1503-06. Specifically, in the present context, comparative case fatality rates refer to death rates of patients with vs without mental disorders among those with sepsis, while mortality rates (as in reference 2) refer to death rates due to infection/sepsis among the populations at risk (e.g., the population of patients with mental disorders vs the general population).

Rather, the findings of lower case fatality among patients with mental disorders who developed septic shock in the present study are consistent with the majority of prior reports on this topic (references 13-16) in hospitalized patients with sepsis (as well as those with septic shock in some of the latter studies), while contrasting 2 other studies (references 11-12).

Notably, the premise of the present study as stated in the Abstract Background is to examine whether the reported findings of higher mortality rates due to infection among patients with mental disorders are similarly increased for septic shock, using the later as study hypothesis. As noted above on terminology, the usual approach to such latter study would be to conduct an investigation on the death rates due to septic shock among the population with the selected mental disorders vs in the general population. Instead, the present study design has examined case fatality among septic shock patients with vs without mental disorders. This later approach cannot answer authors' hypothesis as stated. Similar mix occurs in authors' justification for the present study in the last paragraph of the Introduction section, stating "To the best of our knowledge… no study accounting for the most relevant confounding variables has determined whether septic shock is associated with excess mortality in patients with SMI". As applied to the question examined in present manuscript, the issue is rather whether SMI is associated with excess mortality among patients with septic shock (which is indeed the manuscript's title). The Background and Conclusions of the Abstract, the relevant text of the last paragraph of the Introduction section, and the Conclusions section of the main manuscript should be revised accordingly to enhance the clarity of the study question and interpretation of the study findings. 

Instead, a key inference of the findings of the present study is that the higher mortality rate due to infection/sepsis among patients with mental disorders compared to the general population appears to be due to the increased risk of infection/sepsis among the former (as noted by the authors in the Introduction) and not due to greater case fatality once they develop sepsis. This finding has major implications on future focus of resource allocation for efforts to improve life expectancy in patients with mental disorders. 

Adding this latter inference to the Discussion will better clarify the study implications.

2. In the following sentence of the Discussion (referencing comment #1), the authors hypothesize on the causes of the higher mortality rates due to infection among patients with mental disorders (as in reference 2). As noted in #1 above, such postulated factors are not relevant for comparison of the present study with studies on morality rates but would be potentially applicable for comparison of the present study with others on case fatality of septic patients with vs without mental disorders.

The hypothesized greater delays in sepsis care and triage to ICU among septic patients with mental disorders vs the general population are plausible and are supported by the discussed and refenced background. However, such greater delays in care/ICU triage among septic with mental disorders were also likely present in hospitals in the present study and the authors do not provide data to suggest otherwise. 

Indeed, it is likely that such postulated delays in care/ICU triage among septic patients with mental disorders were also prevalent in prior studies on this topic (references 11-16) and thus would have been expected to lead consistently to higher case fatality among the groups with mental disorders, and a similar finding would have been expected in the present study. 

Instead, a key finding of the present study (and most the prior ones; references 13-16) is that some factors unique to patients with mental disorders (e.g., possibly baseline immune dysfunction, likely leading to a different, more protective, response to infection) have not only negated the adverse prognostic effects of the postulated greater delays in care/ICU triage in septic patients with mental disorders (which may have resulted in similar case fatality between the groups), but were associated with markedly lower case fatality among these patients. Such magnitude of effect estimate is remarkable and quite uncommon. No other alternative confounders, not used for modeling in the present study, appear likely to explain the study findings.

These latter issues need to be discussed as another key implication of the study findings.

3. Studies describing the prognostic implications of specific conditions or socioeconomic factors in sepsis report nearly universally findings of traits associated higher case fatality. Such findings serve as background for subsequent mechanistic studies geared to inform efforts to reduce outcome disparities.

The present study reports instead on patient groups with lower case fatality in sepsis compared to the general population. What is the clinical/research relevance of reporting such "negative" findings for sepsis care? Although the authors touch briefly and indirectly on this topic at the end of the Conclusions section, further, more explicit, discussion is warranted to address the related implications of the present study.

4. The key exposure in the present study (mental disorders) can be misclassified due to use of ICD codes (or other codes), which can affect not only specific characteristics within groups, such as obesity. Such misclassification could have affected reported effect estimates. This should be added to the study limitations.

5. The authors did not report on processes of care for sepsis, which may have differed across compared groups and thus could have led to residual confounding in modeled effects. This should be added to the study limitations.

Minor comments 

1. The terms 30-day hospital morality, 90-day hospital mortality, and 1-year hospital morality should be revised by removing the word "hospital". 

2. The Methods subsection title "Procedures' should be changed to "Exposures".

3. The covariate termed "delay to ICU admission" (modeled as ≤1 vs > 1 day) suggests essentially lack of timely triage to ICU. However, the authors do not provide data on patients' pre-ICU clinical status to allow such inference. The term should be changed in Tables and manuscript to time to ICU admission since hospital admission or similar term to avoid misinterpretation.

4. The Results narrative indicates that unadjusted 90-day mortality was lower among patients in all mental disorders groups. However, per Table 1, there was no statistically significant difference in 90-day mortality in between septic shock patients with vs without major depressive disorder, with similar lack of statistical significance for 30- and 1-year mortality. The text should be revised. 

Reviewer #2: This is an interesting study on the association between severe mental illness and 90-day hospital mortality in septic shock patients in France. However, there are quite a few major issues needing attention.

1) In the methods and findings in the abstract, it says "Compared to matched controls, before and after adjustment, ..." but then the presented results - percentages and HR, are they adjusted or unadjusted? It's not clear.

2) The presentation of 6 models is difficult to follow, confusing and redundant. Basically, we only need one ultimate and fully adjusted model as primary analysis, and the rest can go to supplementary information including Figure 2. We want to see the full details of this primary and fully adjusted analysis, not only the main outcomes between those with and without SMI but also the HRs for all those variables adjusted the cox model including all the demorgraphics, case-mix, risk factors and etc. In the way, we may find some clue to explain the findings of this study.

3) Missing data. What are missing rates for patients' data, variable by variable? How was the missing data issue dealth with in the analyses? The authors didn't mention the missing data at all in the paper, which is inadequate.

4) There are miss-matches between case and control, which may help to explain the findings. For table 1 and 2, the comparisons were done using the standardized difference, however it's only one way for comparison. More often we use p-values for comparison. Could authors please use appropirate statistical tests to get p-values for comparison between case and control for the variables? The p-values and SD can be presented side by side. We need to go to the details of the matching proces to see whether it's properly done.

5) The authors concluded that "In contrast to the excess mortality from infection observed in patients with SMI, our findings suggested improved outcome from septic shock in the ICU". However, the results are a bit difficult to understand. The interpretation and explanation in the discussion are mainly from some theories and references but not comprehensive and basically not convincing. Maybe worth going back to the study data and investigate the differences in characteristics of the patients with and without SMI and see if it can help the interpretation.

6) 90-day hospital mortality. Does it mean those patients died at hospital by 90 days? What if a patients was discharged before 90-days, I assume there are many, but the subsequently died. Do you have this information and how it's recorded. Basically I'd like to know what is the difference between 90-day hospital mortality and 90-day mortality, and many studies used the latter one. 

Reviewer #3: This is a review of the manuscript "Severe mental illness is associated with a better prognosis in septic shock: results from 187,587 hospitalizations in France" submitted for publication in PLOS Medicine. This is a very interesting manuscript, which addresses an important question, with potential important therapeutic implications. There is much to like about this manuscript. The results are presented clearly, the methods are sound, the discussion follows well from it, and the manuscript is very well written. Below are my comments and suggestions:

Main comments:

1/ As stated by the authors, the reasons for lower septic shock's related mortality in patients with severe mental illnesses (SMI) than in other patients may possibly be linked to their exposures to specific psychotropic medications. I think that the interest in this manuscript would be even more pronounced if secondary exploratory analyses could test the moderating effects of main categories of psychotropic medications in these associations. A complementary approach would be to compare mortality rates between the patients taking psychotropic medications versus not. 

2/ Are diagnoses of psychiatric disorders mutually independent or is there a possibility of overlaps? If it is the case, I would present the correlations across disorders. 

3/ I think that the sentence "This hypothesis has been recently reinforced during the COVID-19 pandemic, during which fluoxetine [46] (an antidepressant) and chlorpromazine [47](an antipsychotic) were suggested to have beneficial effects" could be expanded further. 

Specifically, a recently published SARS-CoV-2 animal model showed partly independent antiviral and anti-inflammatory effects of fluoxetine (PMID: 36362409), in line with several observational studies (PMID: 36233753; PMID: 35241663; PMID: 36380766). A candidate mechanism, which is shared by several psychotropic medications and is supported by several preclinical (PMID: 34608263) and observational studies (PMID: 36233753; PMID: 35241663; PMID: 35995770; PMID: 34050932), is the Functional Inhibition of Acid Sphingomyelinase (FIASMA). Finally, several RCTs and observational interventional studies have also found evidence of efficacy of fluvoxamine at a daily dose of 200mg or more against COVID-19 among outpatients with COVID-19 (PMID: 33180097; PMID: 34717820; PMID: 35385087) and ICU COVID patients (PMID: 34719789). I think that the inclusion of these data may enrich the discussion. 

4/ Another potential implication of these results is that the frequently observed discontinuation of psychotropic medications in ICUs should be carefully discussed given the risks of relapse of the psychiatric disorder as well as their potential benefits on mortality in the context of septic shock.

[LINK]

---

## [Decision Letter · Decision Letter 2]

7 Feb 2023

Dear Dr. Boyer,

Thank you very much for re-submitting your manuscript "Association of severe mental illness and septic shock case fatality rate in patients admitted to the intensive care unit: a national population-based cohort study" (PMEDICINE-D-22-03397R2) for review by PLOS Medicine.

I have discussed the paper with my colleagues and the academic editor and it was also seen again by 3 reviewers. I am pleased to say that provided the remaining editorial and production issues are dealt with we are planning to accept the paper for publication in the journal.

[LINK]

We look forward to receiving the revised manuscript by Feb 15 2023 11:59PM.   

Sincerely,

Philippa Dodd, MBBS MRCP PhD

PLOS Medicine

plosmedicine.org

Requests from Editors:

GENERAL

Thank you for your detailed and considerate responses to previous editor and reviewer comments, which we appreciate and accept. Please see below for further minor revisions which we require that you address in full.

AUTHOR SUMMARY

Thank you for including an author summary which reads very nicely – please see reviewer comments below for suggested revisions.

FIGURES

To make your figures more accessible to those with colour blindness, please consider avoiding the use of red (or green).

SOCIAL MEDIA

To help us extend the reach of your research, please provide any Twitter handle(s) that would be appropriate to tag, including your own, your coauthors’, your institution, funder, or lab. Please detail any handles you wish to be included when we tweet this paper, in the manuscript submission form when you re-submit the manuscript.

COMMENTS FROM THE ACADEMIC EDITOR

Please see attachment

Comments from Reviewers:

Reviewer #1: The authors have addressed systematically the vast majority of the comments, with extensive revision of the manuscript. Few issues remain.

1. The clinical/research relevance of reporting a study on a patient group with lower (rather than higher) case fatality among septic patients was addressed only in a narrow fashion (e.g., need to exercise care when considering discontinuation of psychotropics on ICU admission). 

Specifically, a key long-term goal of identifying patient groups with higher case fatality in sepsis than that in the general population (as commonly reported) is (beyond providing clinicians data for prognostication and other bedside decision-making) to identify the mechanisms underlying the outcome differences and, critically, modifiable mechanisms that can serve as targets for interventional approaches geared to reduce the outcome disparity of the affected group in reference to the general population. 

On the other hand, the present study demonstrates an unexpectedly lower case fatality in a patient group (severe mental disorders) that would be expected to have higher case fatality, where the studied group is, critically, well-documented to have immune dysfunction across multiple domains at baseline. Thus, a key implication of this study is that future work to characterize potential differences in response to infection among patients with and without severe mental disorders across key domains of the immune system may identify potential targets for therapeutic interventions to reduce short-term mortality in the general population. Adding these implications to the Discussion can provide readers with broader context for the study's findings.

2. The authors have added, as recommended, to the study limitations the potential of miscoding of mental disorders and lack of data on processes of care across examined groups in their cohort. Addressing some key implications of these limitations can add further context to the study findings. Specifically, misclassification of mental disorders would be expected to blur the differences between groups and thus diminish outcome differences between septic shock patients with and without mental disorders; this would suggest that the study's findings may represent possible underestimation of the magnitude of the better outcomes observed among patients with mental disorders. Similarly, as related to unknown and potentially different processes of care between septic shock patients with and without severe mental disorders, patients with mental disorders are well-documented to receive poorer quality of healthcare, and stigma, stereotyping, and negative attitudes towards these patients by clinicians appear prevalent. Such care differences would be not be expected, however, to result in better outcomes of septic patients with mental disorders; indeed, as noted for potential miscoding of mental disorders, such potential differences in care processes would suggest that the study's findings may represent possible underestimation of the magnitude of the better outcomes observed among patients with mental disorders. 

3. In the Abstract, under Background, the revised last part of the first sentence is confusing (repeated use of "associated"). Revising that part to read "but whether SMI is associated with higher or lower case fatality rates (CFRs) among infected patients remains unclear" would improve clarity and focus.

4. In the Author Summary, under the section "What do these findings mean?" I would suggest including a brief notation that the findings also indicate that the excess mortality from sepsis (as noted under "Why was this study done?") is due to an increased risk of sepsis/infection among patients with severe mental disorders, but not due to increase case fatality among septic patients. 

Reviewer #2: Thanks authors for their great effort to improve the manuscript. All my comments were well addressed in a professional way. I am satisfied with the response and revision. No further issues needing attention.

Reviewer #3: The authors did a great job in responding to my comments. 

I thank them and congratulate them for this well-designed and important study.

[LINK]

---

## [Editor Report · Decision Letter 3]

16 Feb 2023

Dear Dr Boyer, 

On behalf of my colleagues and the Academic Editor, Professor Charlotte Hanlon, I am pleased to inform you that we have agreed to publish your manuscript "Association of severe mental illness and septic shock case fatality rate in patients admitted to the intensive care unit: a national population-based cohort study" (PMEDICINE-D-22-03397R3) in PLOS Medicine.

Please be reminded to include your twitter handles (@MarcLeone8 @GuillaumeFond @LakbarInes @univamu @aphm_actu) in the manuscript submission form.

PRESS

Best wishes,

Pippa 

Philippa Dodd, MBBS MRCP PhD 

PLOS Medicine